# CANIFE: CRAFTING CANARIES FOR EMPIRICAL PRIVACY MEASUREMENT IN FEDERATED LEARNING

**Samuel Maddock**[*]
University of Warwick

**Alexandre Sablayrolles**
Meta AI

**Pierre Stock**
Meta AI

## ABSTRACT

Federated Learning (FL) is a setting for training machine learning models in distributed environments where the clients do not share their raw data but instead send model updates to a server. However, model updates can be subject to attacks and leak private information. Differential Privacy (DP) is a leading mitigation strategy which involves adding noise to clipped model updates, trading off performance for strong theoretical privacy guarantees. Previous work has shown that the threat model of DP is conservative and that the obtained guarantees may be vacuous or may overestimate information leakage in practice. In this paper, we aim to achieve a tighter measurement of the model exposure by considering a realistic threat model. We propose a novel method, CANIFE, that uses *canaries*—carefully crafted samples by a strong adversary to evaluate the empirical privacy of a training round. We apply this attack to vision models trained on CIFAR-10 and CelebA and to language models trained on Sent140 and Shakespeare. In particular, in realistic FL scenarios, we demonstrate that the empirical per-round epsilon obtained with CANIFE is $4 - 5\times$ lower than the theoretical bound.

## 1 INTRODUCTION

Federated Learning (FL) has recently become a popular paradigm for training machine learning models across a large number of clients, each holding local data samples (McMahan et al., 2017a). The primary driver of FL's adoption by the industry is its compatibility with the "privacy by design" principle, since the clients' raw data are not communicated to other parties during the training procedure (Kairouz et al., 2019; Huba et al., 2022; Xu et al., 2022). Instead, clients train the global model locally before sending back updates, which are aggregated by a central server. However, model updates, in their individual or aggregate form, leak information about the client local samples (Geiping et al., 2020; Gupta et al., 2022).

Differential Privacy (DP) (Dwork et al., 2006; Abadi et al., 2016) is a standard mitigation to such privacy leakage. Its adaptation to the FL setting, DP-FEDAVG (McMahan et al., 2017b), provides user-level guarantees by adding Gaussian noise to the aggregated clipped model updates received by the server. In practice, training with strong privacy guarantees comes at the expense of model utility (Bassily et al., 2014; Kairouz et al., 2019), notwithstanding efforts to close this gap, either with public pre-training and partial model updates (Xu et al., 2022), accountants with better compositionality properties (Mironov, 2017) or DP variants such as DP-FTRL (Kairouz et al., 2021).

Hence, it is common in practical deployments of DP-FL to train with a high privacy budget $\varepsilon$ resulting in loose privacy guarantees (Ramaswamy et al., 2020). Such large privacy budgets often provide vacuous guarantees on the information leakage, for instance, against membership inference attacks (Mahloujifar et al., 2022). Encouragingly, recent work has shown that the information recovered in practice using state-of-the-art attacks is less than what theoretical bounds may allow (Nasr et al., 2021). This suggests that DP is conservative and that a tighter measurement of the model exposure may be achieved by considering more realistic threat models.

In this paper, we propose to complement DP-FL training with a novel attack method, CANaries In Federated Environments (CANIFE), to measure empirical privacy under a realistic threat model. We assume that a rogue client wants to reconstruct data samples from the model updates. To make its job

---

[*]Work done during an internship at Meta.

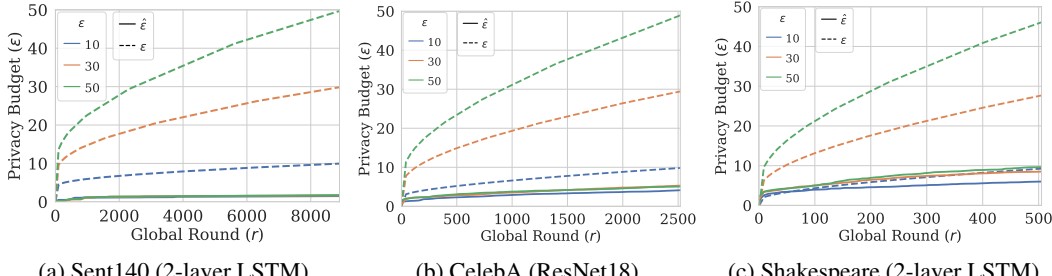

(a) Sent140 (2-layer LSTM)   (b) CelebA (ResNet18)   (c) Shakespeare (2-layer LSTM)

Figure 1: Empirical privacy measurements over the course of FL training for LEAF benchmarks Sent140, CelebA and Shakespeare with $\varepsilon \in \{10, 30, 50\}$. We observe a notable gap between the theoretical $\varepsilon$ obtained with DP-FEDSGD and the empirical $\hat{\varepsilon}$ obtained with CANIFE.

easier, this adversary is allowed to craft an outlier training sample, the *canary*. The training round proceeds normally, after which the rogue client performs a statistical test to detect the canary in the global noisy model update provided to the server by any secure aggregation protocol (see Figure 2). Finally, we translate the attack results into a per round measure of empirical privacy (Jagielski et al., 2020; Nasr et al., 2021) and propose a method using amplification by subsampling to compute the empirical privacy incurred during training as depicted in Figure 1 for standard FL benchmarks.

Critically, our privacy attack is designed to approximate the worst-case *data sample*, not the worst-case *update vector*. The rogue client seeks to undermine the privacy guarantee by manipulating its input, which is consistent with FL environments using secure sandboxing to protect the integrity of the training process (Frey, 2021). We additionally model the server as the honest party, not allowing it to poison the global model in order to reconstruct training samples, in contrast with a recent line of work (Fowl et al., 2021; Boenisch et al., 2021; Wen et al., 2022; Fowl et al., 2022).

In summary, our contributions are as follows:

- We propose CANIFE (Section 3), a novel and practical privacy attack on FL that injects crafted *canary* samples. It augments the standard DP-FL training with a tight measure of the model's privacy exposure given a realistic yet conservative threat model.
- CANIFE is compatible with natural language and image modalities, lightweight and requires little representative data and computation to be effective. As a sanity check, we demonstrate that CANIFE tightly matches DP guarantees in a toy setup (Section 4.1) before exploring how it behaves in the federated setting (Section 4.2).
- Our work highlights the gap between the practical privacy leakage and the DP guarantees in various scenarios. For instance, on the CelebA benchmark, we obtain an empirical measure $\hat{\varepsilon} \approx 6$ for a model trained with a formal privacy guarantee of $\varepsilon = 50$.

## 2 BACKGROUND

### 2.1 DIFFERENTIAL PRIVACY

Differential Privacy (Dwork et al., 2006; Dwork & Roth, 2014) defines a standard notion of privacy that guarantees the output of an algorithm does not depend significantly on a single sample or user.

**Definition 1** (Differential Privacy). *A randomised algorithm* $\mathcal{M} \colon \mathcal{D} \to \mathcal{R}$ *satisfies* $(\varepsilon, \delta)$-*differential privacy if for any two adjacent datasets* $D, D' \in \mathcal{D}$ *and any subset of outputs* $S \subseteq \mathcal{R}$,

$$\mathbb{P}(\mathcal{M}(D) \in S) \leq e^{\varepsilon}\mathbb{P}(\mathcal{M}(D') \in S) + \delta.$$

The privacy parameter $\varepsilon$ is called the *privacy budget* and it determines an upper bound on the information an adversary can obtain from the output of an $(\varepsilon, \delta)$-DP algorithm. The parameter $\delta$ defines the probability of failing to guarantee the differential privacy bound for any two adjacent datasets. In this work, we are interested in *user-level* differential privacy which takes $D$ and $D'$ to be *adjacent* if $D'$ can be formed by adding or removing all samples associated with a single user from $D$.

Standard DP results state that the privacy budget $(\varepsilon, \delta)$ accumulates (roughly) in proportion to the square root of the number of iterations. Advanced privacy accountants leverage the uncertainty due to random sampling (Mironov, 2017; Wang et al., 2019b; Gopi et al., 2021). We use the Rényi Differential Privacy (RDP) accounting implemented in the Opacus library (Yousefpour et al., 2021).

## 2.2 Private Federated Learning

A standard FL protocol, such as FEDAVG (McMahan et al., 2017a) computes a weighted average of the model updates from clients before performing a gradient descent step on the global model. Other variants exist to deal with common optimization problems in the federated setting such as convergence speed (Wang et al., 2019a), heterogeneity (Karimireddy et al., 2020a;b), reducing communication bandwidth (Alistarh et al., 2017) and adding momentum (Reddi et al., 2020). In this work, we focus on DP-FEDSGD, a private extension of FEDAVG (McMahan et al., 2017b). At each round, the selected clients compute their clipped model update $u_i$ and transmit it to the server, which aggregates the model updates and adds Gaussian noise:

$$\tilde{u} = \sum_i u_i + \mathcal{N}(0, \sigma^2 I_d).$$

In practice, federated algorithms rely on Secure Aggregation (SegAgg) protocols to aggregate each client update without revealing any individual $u_i$ to the server or to other participants (Bonawitz et al., 2017; Bell et al., 2020). In a TEE-based SecAgg (Huba et al., 2022), a trusted execution environment (TEE) aggregates the individual model updates and calibrated DP noise before handing over the noisy model update $\tilde{u}$ to the server. These specific models are orthogonal to our work and we assume from now on that the server and clients participate in a TEE-based SecAgg protocol. We discuss our threat model with regards to our attack in Section 3.1.

## 2.3 Attacks & Empirical Privacy

**Centralized and FL Attacks.** There is a vast body of literature on attacking models trained in the centralised setting. For example, membership inference attacks (MIA) attempt to distinguish whether a sample was present in the training set given only the trained model (Shokri et al., 2017; Sablayrolles et al., 2019). Others attacks consider the more difficult problem of reconstructing entire training samples from a trained model, often using (batch) gradient information (Yin et al., 2021; Jeon et al., 2021; Balle et al., 2022). Since model updates $u_i$ are essentially just aggregated gradients, it is natural that FL updates may leak private information as well. Nasr et al. (2019) show that it is possible to perform both passive and active membership-inference attacks in the federated setting. Other works such as that of Fowl et al. (2021) and Wen et al. (2022) have designed attacks on model updates which allow for the reconstruction of training samples used in federated training. However, they assume that the server can poison the global model, whereas we assume an honest server.

**Canaries.** The notion of canary samples usually refers to natural data samples used to measure memorization in large language models (Carlini et al., 2019; 2021; Thakkar et al., 2020; Shi et al., 2022). For instance, Parikh et al. (2022) propose to reconstruct canaries inserted in training data and Stock et al. (2022) insert canaries to track and quantify the information leakage when training a causal language model. In all prior work, the canary is either a sample from the training set or a handcrafted instance such as "My SSN is 123-45-6789". In contrast, CANIFE provides an explicit method for crafting canary samples that are as adversarial as possible within the given threat model (see Appendix E for canary samples) to obtain tight measurement of the model's exposure. Moreover, the proposed method applies to any training modality allowing to backpropagate in the sample space (in particular, pixels for images and tokens for natural language).

**Empirical Privacy.** The proposed approach departs from existing attacks in the FL setup. For instance, Jayaraman & Evans (2019) and Jagielski et al. (2020) have derived empirical measures of privacy through attacks and often shown gaps between the empirically measured and theoretical privacy. More recently, Nasr et al. (2021) study a range of membership-inference attacks by varying adversary's powers. We argue that the threat model of many of these attacks is often too permissive relative to what a realistic adversary can achieve in the federated setting. For example, attacks in Nasr et al. (2021) assume knowledge of the other samples in the dataset.

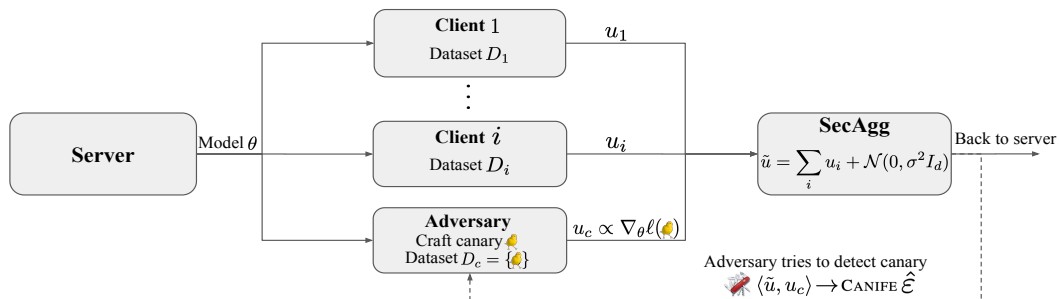

Figure 2: Illustration of the proposed CANIFE method for one training round. The adversary is a rogue client that crafts a training sample resulting in an extremely out-of-distribution model update $u_c$ and performs a membership inference attack on the (public) aggregated noisy model update $\tilde{u}$ provided to the server by SecAgg. Attack results are converted to empirical privacy guarantees $\hat{\varepsilon}$ for this round. We then compound these guarantees over the course over the training as in Section 3.4.

## 3 METHODOLOGY

The adversary wants to design a canary sample $z$ to measure the empirical privacy of a given FL training round under a realistic threat model defined in Section 3.1. We view this problem as a membership inference game where a rogue client crafts a canary $z$ specific to the global model at round $r$ (Section 3.3) and uses this canary to produce an extremely out-of-distribution model update. After this, the server proceeds as usual, aggregating model updates via any secure aggregation primitive and adding noise. At the end of the round, the adversary attempts to detect the presence of the canary in the aggregated noisy model update $\tilde{u}$ (Section 3.2) and computes a per-round empirical privacy measure. These per-round empirical privacy guarantees are compounded over the whole course of the training to give a final empirical privacy guarantee as explained in Section 3.4.

### 3.1 THREAT MODEL

We assume an *honest-but-curious* threat model where the clients and the server do not deviate from the training protocol (Bonawitz et al., 2017; Kairouz et al., 2019; Huba et al., 2022). In particular, the clients cannot directly manipulate gradients nor the final model updates from the local training and the server cannot modify the model maliciously to eavesdrop on the clients.

To craft the canary for a given training round, we assume that the adversary — a rogue client — has access to (1) the current server-side public model and to (2) emulated *mock clients*. Mock clients are created using a public set of samples called the *design pool* whose distribution is similar to that of the clients. We show in Section 4 that the adversary is able to design a strong and robust canary sample even with a small amount of public data and under mild assumptions about the true data distribution. We also demonstrate that the computational resources required to design the canary are small, which means that even adversaries under resource constraints are able to launch such attacks.

We argue such a threat model is realistic and discuss practical limitations that influence the adversary's success in Section 3.4: our goal is to design an attack that is as adversarial as possible (in order to derive a worst-case measure of empirical privacy) under a set of reasonable assumptions.

### 3.2 CANARY DETECTION

The adversary's objective is to craft a canary $z$ such that the resulting model update is extremely out-of-distribution. More precisely, we require that the canary gradient $\nabla_\theta \ell(z)$ is orthogonal to all individual clipped model updates $u_i$ in the current round: $\langle u_i, \nabla_\theta \ell(z) \rangle = 0$. We demonstrate that this allows the adversary to detect the presence of the canary by separating two Gaussian distributions.

The server trains a model parameterized by $\theta \in \mathbb{R}^d$ with a certain level of DP noise $\sigma$ across a population of clients each holding a local dataset $D_i$. Since the rogue client holds a dataset that

only contains the canary $D_c = \{z\}$, its clipped model update will be proportional[1] to the canary gradient: $u_c \propto \nabla_\theta \ell(z)$. Recall that the aggregated private model update $\tilde{u}$ is formed by the noisy sum of individual clipped model updates: $\tilde{u} = \sum_i u_i + \mathcal{N}(0, \sigma^2 I_d)$. Then, as a counter-factual, if the rogue client does not participate in the training round:

$$\langle \tilde{u}, u_c \rangle \propto \sum_i \underbrace{\langle u_i, \nabla_\theta \ell(z) \rangle}_{=0 \text{ by design}} + \underbrace{\langle \mathcal{N}(0, \sigma^2 I_d), \nabla_\theta \ell(z) \rangle}_{=\mathcal{N}(0, \sigma^2 \|\nabla_\theta \ell(z)\|^2)}.$$

Hence, $\langle \tilde{u}, u_c \rangle$ follows a one-dimensional zero-mean Gaussian with variance $\sigma_c^2$, where $\sigma_c$ accounts for the proportionality factor. Similarly, if the rogue client participates in the training round, $\langle \tilde{u}, u_c \rangle$ follows a one-dimensional Gaussian with the same variance $\sigma_c^2$ centered at $\|u_c\|^2$. Thus, the membership inference game is reduced to separating two Gaussian distributions centered at 0 and $\|u_c\|^2$ respectively. Note that the result is unchanged up to a fixed scaling factor if the client updates $u_i$ are weighted by their local dataset size as in (McMahan et al., 2017b). The testing approach is described in full in Algorithm 1 and involves computing the attack score $s_r := \langle \tilde{u}, u_c \rangle$. We derive a connection with the likelihood ratio test in Appendix A.

### 3.3 CANARY DESIGN

At a given round $r$, the rogue client is given the current server-side model parameterized by $\theta$. Given a set of heldout clipped model updates $\{u_i\}$, it creates the canary by minimizing:

$$\mathcal{L}(z) = \sum_i \langle u_i, \nabla_\theta \ell(z) \rangle^2 + \max(C - \|\nabla_\theta \ell(z)\|, 0)^2. \tag{1}$$

Recall that $\nabla_\theta \ell(z)$ denotes the gradient of the network's parameters with respect to its training loss $\ell$ when forwarding the canary $z$ through the network. The first loss term is designed to make the canary gradient $\nabla_\theta \ell(z)$ orthogonal to the set of heldout model updates while the second term enforces the canary norm is not smaller than some constant, that we set to the clipping constant $C$ of DP-FL. This ensures $\|u_c\|^2 = C^2$ and for simplicity we fix $C = 1$. In Appendix D, we provide experiments that show that choosing the gradient norm constant too large (i.e., much larger than $C$) has a detrimental effect on optimization.

Using an automatic differentiation framework such as PyTorch (Paszke et al., 2019), we compute $\nabla_z \mathcal{L}(z)$ and perform stochastic gradient descent directly in the sample space as described in Algorithm 1. (Recall that the model parameters $\theta$ are fixed during the optimization procedure.) Computing $\nabla_z \mathcal{L}(z)$ is straightforward for continuous data, such as images, as we can simply backpropagate in the pixel space. For language models, the problem is more complex as the input is a sequence of discrete tokens. Hence, we leverage the work of Guo et al. (2021), who use the Gumbel-Softmax distribution (Jang et al., 2016) to forge adversarial language examples. This allows them to use gradient-based methods by optimising a probability distribution over each token in a sequence.

We investigate various methods to initialize the canary, including starting from a random training sample or random pixels or tokens. Depending on the task at hand, we might need to fix a target for the canary. For instance, for image classification, we need to assign the canary to a class in order to be able to compute $\nabla_\theta \ell(z)$. We investigate various canary initialization and target choice strategies experimentally in Section 4. We also investigate other optimization considerations for designing the canary such as slightly modifying the loss in Appendix D.

**Adversarial Examples.** We can view the canary $z$ as "adversarial" in the sense that it should be extremely out-of-distribution to get a worst-case measure of privacy. This is different from the unrelated notion of *adversarial examples* which typically constrain the changes of a sample to be imperceptible to humans (Biggio et al., 2013). In our setup, we do not impose this constraint as we wish to encompass the realistic worst-case of an extreme out-of-distribution sample.

### 3.4 MEASURING EMPIRICAL PRIVACY

The CANIFE attack is carried out over a certain number ($n$) of fake rounds where the server does not update its global model but where the pool of selected (regular) clients differs every time. Hence,

---

[1]We assume that the client's local optimizer is SGD with no momentum (McMahan et al., 2017b).

---

**Algorithm 1** CANIFE attack by a rogue client

---

**Input:** Design pool $D_{\text{pool}}$, Design iterations $T$, Canary learning rate $\beta$, Global model $\theta$
 1: Form mock clients from the design pool $D_{\text{pool}}$
 2: For each mock client $i$, compute the clipped model update $u_i$
 3: Initialise the canary $z_0$               ▷ See Section 3.3
 4: **for** $t = 1, \ldots, T$ **do**
 5:    $\mathcal{L}(z_t) \leftarrow \sum_i \langle u_i, C \cdot \nabla_\theta \ell(z_t) \rangle^2 + \max(C - ||\nabla_\theta \ell(z_t)||, 0)^2$   ▷ Canary optimization loss
 6:    Compute $\nabla_{z_t} \mathcal{L}(z_t)$         ▷ Gradient of the canary loss w.r.t $z_t$
 7:    $z_{t+1} \leftarrow z_t - \beta \cdot \nabla_{z_t} \mathcal{L}(z_t)$         ▷ For NLP, see Section 3.3
 8: **end for**
 9: Compute $u_c$           ▷ Model update with $D_c = \{(z_T, y_c)\}$
10: **return** $s_r \leftarrow \langle \tilde{u}, u_c \rangle$      ▷ $\tilde{u}$ is deemed public after the round has finished

---

CANIFE has no influence on the model's final accuracy. The adversary crafts a single canary and inserts it into every fake round with probability $1/2$. Once the $n$ attack scores are computed, the adversary deduces the performance of the attack at a calibrated threshold $\gamma$. Building on previous work (Jagielski et al., 2020; Nasr et al., 2021), we compute the empirical privacy guarantee $\hat{\varepsilon}$ based on the False Positive Rate (FPR) and False Negative Rate (FNR) of the attack at the threshold $\gamma$ as

$$\hat{\varepsilon} = \max \left( \log \frac{1 - \delta - \text{FPR}}{\text{FNR}}, \log \frac{1 - \delta - \text{FNR}}{\text{FPR}} \right).$$

Our measure differs slightly from that of (Nasr et al., 2021) as our attack measures privacy at a single round of training. Thus, $\hat{\varepsilon}$ is a *per-round* privacy measure which we denote as $\hat{\varepsilon}_r$ and we compare this to a per-round theoretical $\varepsilon_r$. In order to deduce an overall empirical epsilon $\hat{\varepsilon}$ we convert the per-round $\hat{\varepsilon}_r$ into a noise estimate $\hat{\sigma}_r$ under amplification by subsampling. We then compose for $s$ steps with an RDP accountant, until we perform the next attack and update our $\hat{\sigma}_r$. To provide a worst-case measure we choose a threshold that maximises $\hat{\varepsilon}$ under our attack. We note that $n$ determines bounds on $\hat{\varepsilon}_r$ and its CIs. With $n = 100$, $\hat{\varepsilon}_r$ is at most 3.89. Similarly, when computing $\hat{\varepsilon}_r$, we set $\delta = 1/n$. For further accounting details and the algorithm to compute $\hat{\varepsilon}$ see Appendix C.

**Relationship to DP.** Our threat model and canary design procedure have been chosen to ensure that the adversary is as strong as possible but still constrained by realistic assumptions. At least three factors of the design process restrict the adversary in practice:

1. The adversary only has partial knowledge of the model updates in the form of an aggregated noisy sum $\tilde{u}$. Furthermore, when designing a canary, the adversary uses a design pool to mock clients. In our design, we have been conservative by assuming the adversary has access to heldout data that matches the federated distribution by constructing the design pool from the test set of our datasets. In practice, adversaries can form design pools from public datasets that match the task of the target model. See Appendix F for further discussion.

2. The optimization process induces two possible sources of error: the convergence itself and the generalization ability of the canaries to unseen model updates.

3. We calculate the maximum accuracy that can be derived from the attack and use the threshold that maximises $\hat{\varepsilon}_r$. This is conservative, as in practice an adversary would have to calibrate a threshold $\gamma$ on a public heldout dataset. Furthermore, they would not be able to perform $n$ fake rounds with the same canary.

If these practical constraints were not imposed on an adversary, then the privacy measure we derive from our attack would essentially be tight with the DP upper bound. For example, if we could always design a perfect canary $z$ that has gradient orthogonal to all possible model updates $u_i$ then the DP noise $\sigma$ completely determines whether we can distinguish if $z$ is present in $\tilde{u}$ or not. However, in practice, the assumptions listed above make the attack more difficult as we will see in Section 4.1.

## 4 EXPERIMENTS

In this section, we begin by highlighting the power of our attack on the simplest federated setting where participants only have a single sample (Section 4.1). We then show how our attack allows

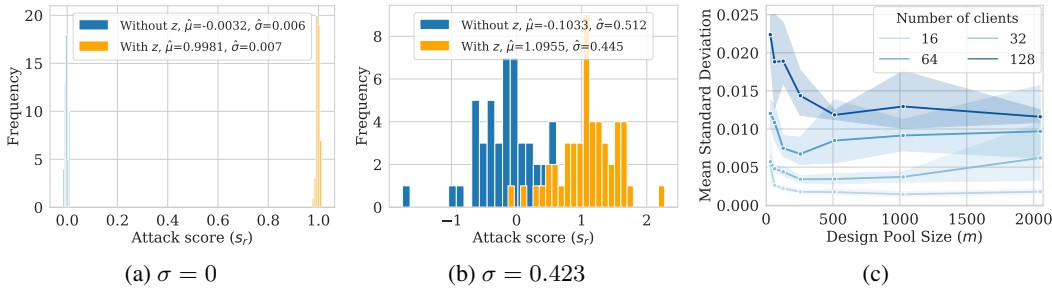

(a) $\sigma = 0$        (b) $\sigma = 0.423$        (c)

Figure 3: Experiments on CIFAR-10 on a toy setup where each client holds a single data sample. (a) Attack histograms without DP noise, showing that our attack completely succeeds. (b) Attack histograms with DP noise: as the DP noise $\sigma$ increases, it becomes harder to separate both histograms. (c) Varying the design pool size and the number of participating clients. The $y$-axis shows the mean standard deviation of the histograms formed from the attack.

us to monitor privacy empirically over a federated training run (Section 4.2) before concluding with an ablation study that shows our canary is robust to various design choices (Section 4.3). We open-source the code for CANIFE design and test to reproduce our results [2].

**Setup.** We evaluate our attack on both image and language models. We utilise LEAF (Caldas et al., 2018) which provides benchmark federated datasets for simulating clients with non-IID data and a varying number of local samples. We study image classification on CIFAR10 (IID) (Krizhevsky et al., 2009) and CelebA (non-IID) (Liu et al., 2015). We train a simple Convolutional Neural Network (CNN) and a ResNet18 model. For our language tasks, we train an LSTM model on non-IID splits of Sent140 (Go et al., 2009) and Shakespeare (McMahan et al., 2017a). For more information on datasets, model architectures and training hyperparameters, see Appendix B. We train, design, and evaluate our attacks using the FLSim framework[3]. All experiments were run on a single A100 40GB GPU with model training taking at most a few hours. We discuss CPU benchmarks for canary design in Section 4.3. For canary design, we use the Adam optimizer (Kingma & Ba, 2014) with learning rate $\beta = 1$ and fix $C = 1$. We form the CANIFE design pool from a LEAF test split, resulting in canaries designed on non-IID mock clients which approximates the training distribution. We have clients perform a single local epoch in all experiments, but see Appendix F for possible extensions to multiple local epochs. From now, we refer to "epoch" as one pass of the federated dataset (in expectation). For privacy accounting, we utilise the RDP accountant with subsampling (Mironov et al., 2019) implemented via the Opacus library (Yousefpour et al., 2021), sampling clients uniformly at each round (i.e., Poisson sampling), see Appendix C for more details.

## 4.1 EXAMPLE ATTACK: CIFAR10

We first investigate the simplest federated setting where each client holds a single sample. It follows that the clipped model update $u_i$ is simply a scaled gradient of the client's single sample. This corresponds to the central setting where the number of clients per round is equivalent to a (central) batch size. In Figure 3, we train a ResNet18 model on CIFAR10 to $60\%$ test accuracy and perform our attack. We design a single canary and have the server perform $n = 100$ mock rounds where the model is frozen, with half having the canary client inserted and half without. We compute attack scores from these $n = 100$ trials and form histograms.

In Figure 3a, we present one histogram of the attack scores on a model that has 64 clients participating per round with no DP ($\varepsilon = \infty$). We use a design pool of $m = 512$ to design the canary. We observe that our attack is quite tight in this non-private setting. Since $\sigma = 0$, we hope that if the canary was designed well, the standard deviation of the histograms would also be close to 0. It turns out that the average standard deviation of the histograms is 0.006. Recall in Section 3.4, we discussed there is inherent error from both the optimization procedure and the fact the server

---

[2]Code available at https://github.com/facebookresearch/canife

[3]https://github.com/facebookresearch/FLSim

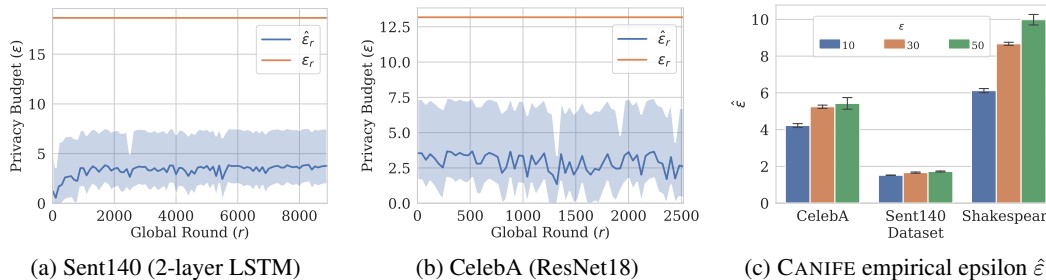

(a) Sent140 (2-layer LSTM)  (b) CelebA (ResNet18)  (c) CANIFE empirical epsilon $\hat{\varepsilon}$

Figure 4: (a) and (b) Monitoring per-round empirical privacy on LEAF benchmarks. Models are trained to $\varepsilon = 50$ with 100 clients per round. CIs for $\hat{\varepsilon}_r$ are compared with the theoretical per-round epsilon $\varepsilon_r$, rounds with an upper CI of $\infty$ do not have CIs displayed. (c) CANIFE empirical privacy incurred during the whole training, averaged over 5 independent runs (see Figure 1 for details).

designs the canary on heldout data but in this case the error is small. In Figure 3b, we display another example attack, this time for a model trained under privacy with a final $\varepsilon = 25$ corresponding to $\sigma = 0.423$ with 64 clients per round. We observe the attack is still fairly tight as the standard deviation of the histograms (average $0.478$) is close to that of the privacy noise $\sigma$.

Finally, we explore how both the design pool size and the number of clients affect the standard deviation of the histograms (Figure 3c). We vary both the number of clients and the design pool size. We train a model without privacy for each combination of parameters until it reaches 60% train accuracy and then attack the model, plotting the average standard deviation of the histograms. We conclude with two observations. First, the attack is much tighter when there is a smaller number of clients participating per round. Second, the size of the design pool has a diminishing effect on reducing the standard deviation. We further explore these in Section 4.3.

## 4.2 MONITORING EMPIRICAL PRIVACY

One main goal of our attack is to provide a lightweight method for monitoring empirical privacy during federated training. We explore this in Figure 4, training ResNet18 on CelebA and a 2-layer LSTM model on Sent140 and Shakespeare. We train to a final $\varepsilon = 50$ and achieve $63.9\%$ test accuracy on Sent140, $89.9\%$ on CelebA and $44.8\%$ on Shakespeare. We carry out the CANIFE attack at a regular interval, freezing the model, designing a single canary, and performing $n = 100$ attack trials with the designed canary before continuing training. This generates scores which are used to compute an empirical per-round privacy measure $\hat{\varepsilon}_r$ as outlined in Section 3.4 (see also Appendix C.3 for further details). The canary sample $z$ is initialised randomly and we explore how this affects optimization in Section 4.3. We note in practice that participants (and the server) would not want to waste rounds with a frozen model, see Appendix F for possible extensions.

We monitor the optimization of the canary loss by computing the *canary health* which measures the percentage improvement of the final canary loss compared to the initial loss. A perfect health of 1 means the canary achieved minimal design loss. As an example, for Sent140, we find that the average canary health across 5 separate training runs is $0.970 \pm 0.098$. Excluding the first 1000 rounds from each run, the average canary health becomes $0.990 \pm 0.01$. Thus, after two model epochs, optimization stabilises and the initial canary loss is reduced by 99%.

In Figure 4a, we display per-round privacy estimates $\hat{\varepsilon}_r$ and their 95% confidence intervals (CIs) for Sent140 trained with $\varepsilon = 50$ and compare to the (constant) theoretical per-round privacy $\varepsilon_r$. We observe $\hat{\varepsilon}_r$ is initially small and grows to an almost constant level within a single epoch and stays there during training. This results in a $4.5\times$ gap between the theoretical $\varepsilon_r$ and $\hat{\varepsilon}_r$ measured by our attack. We obtain similar $4 - 5\times$ gaps for CelebA (ResNet18) in Figure 4b and Shakespeare in Appendix C.5. We compound these per-round estimates $\hat{\varepsilon}_r$ to provide a cumulative measure of privacy $\hat{\varepsilon}$ and display this in Figure 1 for models trained with $\varepsilon \in \{10, 30, 50\}$. Again, there is a significant gap between the final theoretical privacy and the measure derived under CANIFE. The final $\hat{\varepsilon}$ averaged over 5 separate training runs for each tasks are shown in Figure 4c. We note significant gaps between tasks, most notably with Sent140. This is likely due to the small sample

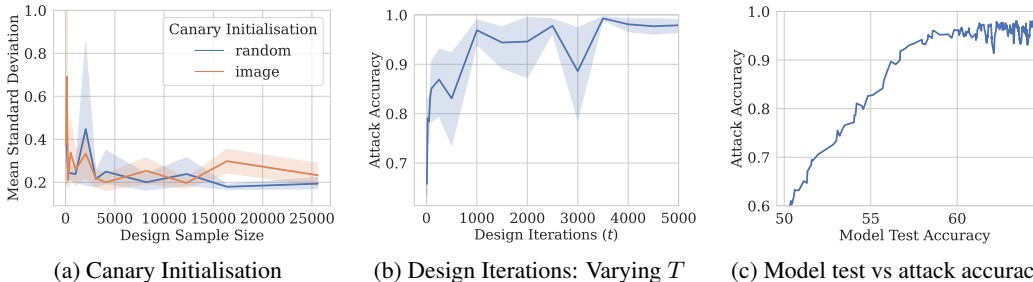

(a) Canary Initialisation  (b) Design Iterations: Varying $T$  (c) Model test vs attack accuracy

Figure 5: Ablations studies for CANIFE **without addition of DP noise**. (a) Canary initialization when varying the design pool size (CelebA, 4-layer CNN). (b) Number of design iterations required to produce robust and well-designed canaries (CelebA, 4-layer CNN). (c) Model vs attack accuracy across 5 runs (Sent140, 2-layer LSTM). Attack accuracy increases as model test accuracy increases.

rate which provides significant privacy amplification. We also observe CANIFE estimates are stable across runs with relatively small standard deviation.

### 4.3 ABLATION STUDY

**Canary Initialisation.** In Figure 5a, we explore how initialising the canary affects the optimization. We consider two initialisation strategies: initialising the canary randomly or initialising the canary as a sample from the design pool (which is then excluded from the design). We observe no significant difference on the average standard deviation of the attack histograms.

**Design Pool Size.** In Figure 5a, we vary the design pool size. We observe there is no significant effect on the average standard deviation of the attack histograms. This confirms what we observed in Figure 3c: the design pool size has diminishing impact on reducing the standard deviation.

**Design Iterations.** In Figure 5b, we explore how the number of design iterations impacts the quality of the canary measured through the calibrated attack accuracy. We observe that just $t = 1000$ iterations are needed to obtain $95\%$ accuracy and that with $t = 3000$ the attack improves to almost $100\%$ accuracy, staying close to constant as $t$ increases further. We additionally benchmarked the average CPU time on an M1 MacBook Air (2020) for a single design iteration. For Sent140, it takes an average of 0.06s per iteration and 0.23s for CelebA (ResNet18). For $t = 2500$, the design procedure took on average 165s and 591s respectively. Hence, the canary design process takes at most 10 minutes on a CPU and only a few minutes with a GPU. Thus our attack is lightweight, requiring only a few thousand design iterations to achieve near-optimal canaries.

**Model Accuracy.** We conclude by noting the accuracy of the model and that of our attack are highly correlated. In Figure 5c, we plot both model test accuracy and calibrated attack accuracy across Sent140 trained without DP. We observe early in training, when model accuracy is low, that the attack accuracy is similarly low. Once the model converges to a sufficiently high test accuracy the attack accuracy is close to $100\%$. In experiments that require comparison across different models (e.g., Figure 3c) we checkpoint to a fixed test accuracy to avoid this confounding effect.

## 5 CONCLUSION

Motivated by the fact that DP is conservative, we consider a more realistic threat model to extract private information about client data when training a model with FL. We introduce CANIFE, a novel method to measure empirical privacy where a rogue client crafts a canary sample that results in an outlier model update. We argue that the difficulty of tracing this malicious model update in the aggregated noisy model update provides a tighter measurement of the model's privacy exposure. We hope this work can benefit practical deployments of DP-FL pipelines by complementing the theoretical bound with an arguably more realistic measure of the privacy leakage.

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

## A    CONNECTIONS TO LIKELIHOOD RATIO TEST

Let us assume that each model update follows a Gaussian distribution $\mathcal{N}(\mu, \Sigma)$. The sum of $k$ model updates then either follows $\mathcal{N}(k\mu, k\Sigma)$ (without the canary) or $\mathcal{N}(k\mu + \nabla\ell(z), k\Sigma)$ (with the canary), recalling that $u_c \propto \nabla\ell(z)$. Then we have

$$p_0(u) = \frac{1}{\sqrt{\det(2\pi k\Sigma)}} \exp\left(-(u - k\mu)^T (k\Sigma)^{-1}(u - k\mu)/2\right)$$

$$p_1(u) = \frac{1}{\sqrt{\det(2\pi k\Sigma)}} \exp\left(-(u - (k\mu + \nabla\ell(z)))^T (k\Sigma)^{-1}(u - (k\mu + \nabla\ell(z)))/2\right)$$

$$= \frac{1}{\sqrt{\det(2\pi k\Sigma)}} \exp\left(-((u - k\mu) - \nabla\ell(z)))^T (k\Sigma)^{-1}((u - k\mu) - \nabla\ell(z))/2\right).$$

We can write the (log) likelihood ratio as

$$\log\frac{p_1(u)}{p_0(u)} = \frac{1}{2}\left((u - k\mu)^T (k\Sigma)^{-1}(u - k\mu) - ((u - k\mu) - \nabla\ell(z)))^T (k\Sigma)^{-1}((u - k\mu) - \nabla\ell(z))\right)$$

$$= \frac{1}{2}\left((u - k\mu)^T (k\Sigma)^{-1}(u - k\mu) - ((u - k\mu) - \nabla\ell(z)))^T (k\Sigma)^{-1}((u - k\mu) - \nabla\ell(z))\right)$$

$$= \nabla\ell(z)^T (k\Sigma)^{-1}(u - k\mu) - \frac{1}{2}\nabla\ell(z)^T (k\Sigma)^{-1}\nabla\ell(z).$$

In particular for the centers of the Gaussian with and without the canary, $u \in \{k\mu, k\mu + \nabla\ell(z)\}$,

$$\log\left(\frac{p_1(u)}{p_0(u)}\right) = \pm\frac{1}{2}\nabla\ell(z)^T (k\Sigma)^{-1}\nabla\ell(z).$$

Maximizing this term will thus help separate the two Gaussians. However, doing this directly is infeasible as it requires to form and invert the full covariance matrix $\Sigma$ in very high dimensions. Instead, we propose to *minimize* $z \mapsto (\nabla\ell(z)^T)\Sigma(\nabla\ell(z))$ as it is tractable and can be done with SGD. Note that for sample model updates $\{u_i\}$ we can estimate the (uncentered) covariance matrix as $\frac{1}{n}\sum_i u_i u_i^T$ and thus

$$(\nabla\ell(z)^T)\Sigma(\nabla\ell(z)) \approx \frac{1}{n}\sum_i \nabla\ell(z)^T (u_i u_i^T)\nabla\ell(z) = \frac{1}{n}\sum_i \langle u_i, \nabla\ell(z)\rangle^2.$$

Which, ignoring constants, is the first term of $\mathcal{L}(z)$ defined in Equation 1. One could alternatively minimise $\langle\nabla\ell(z), \hat{\mu}\rangle^2$ with $\hat{\mu} = \frac{1}{n}\sum_i u_i$. We explore the empirical differences in Appendix D.

## B    DATASETS & MODEL ARCHITECTURES

Here we detail the training setup and model architectures for our experiments. In all experiments we train with DP-FEDSGD and without momentum. We use both client ($\eta_C$) and server ($\eta_S$) learning rates. We train without dropout in all model architectures. In more detail:

- **CIFAR10** is an image classification tasks with 10 classes (Krizhevsky et al., 2009). We train a ResNet18 model (He et al., 2016) on CIFAR10. We form an IID split of $50,000$ train users and $10,000$ test users where each user holds a single sample and thus has a local batch size of 1. We use a client learning rate of $\eta_C = 0.01$ and server learning rate $\eta_S = 1$.

- **CelebA** is a binary image classification task (Liu et al., 2015). We train a ResNet18 model with $11,177,538$ parameters and a simple Convolutional Neural Network (CNN) with four convolutional layers following that used by Caldas et al. (2018) with $29,282$ parameters. We use the standard non-IID LEAF split resulting in $8408$ train users and $935$ test users and a local batch size of 32. We train with a client learning rate of $\eta_C = 0.899$ and a server learning rate of $\eta_S = 0.0797$.

---

**Algorithm 2** Measuring $\hat{\varepsilon}$

---

**Input:** Number of rounds $R$, Privacy parameter $\delta$, Sampling rate $q$, Number of attack scores $n$, Attack frequency $s$
1: **for** $r = 1, \ldots, R$ **do**
2:     **if** $r \mod s = 0$ **then**
3:         Freeze the model $\theta_r$ and use Algorithm 1 to compute attack scores $\{s_i\}_{i=1}^n$
4:         Calculate $\text{FPR}_\gamma$ and $\text{FNR}_\gamma$ from $\{s_i\}$ at each threshold $\gamma \in \{s_1, \ldots, s_n\}$
5:         Compute $\hat{\varepsilon}_r \leftarrow \max_\gamma \left( \log \frac{1 - \delta - \text{FPR}_\gamma}{\text{FNR}_\gamma}, \log \frac{1 - \delta - \text{FNR}_\gamma}{\text{FPR}_\gamma} \right)$
6:         Compute $\hat{\sigma}_r \leftarrow \text{GetNoise}(\hat{\varepsilon}_r, \delta)$          ▷ Estimate one-step noise multiplier
7:         $\hat{\sigma}_{r+i} \leftarrow \hat{\sigma}_r$ for $i \in [1, s]$          ▷ $\hat{\sigma}_r$ is the estimate for rounds $[r, r+s]$
8:     **end if**
9: **end for**
10: **return** $\hat{\varepsilon} \leftarrow \text{GetPrivacy}(\{\hat{\sigma}_r\}; \delta, q)$    ▷ Compose each noise estimate under an RDP accountant with amplification by subsampling using sample rate $q$, see (Mironov et al., 2019)

---

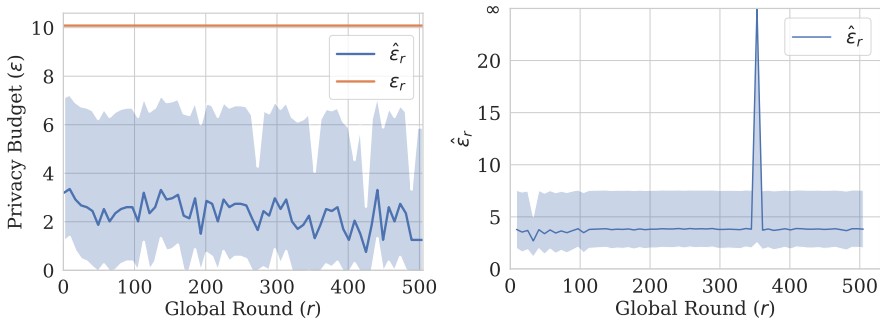

(a) Shakespeare per-round privacy ($\varepsilon = 50$)    (b) Shakespeare per-round privacy ($\varepsilon = \infty$)

Figure 6: Measuring privacy during Shakespeare training

- **Sent140** is a sentiment analysis (binary classification) task (Go et al., 2009). We train a 2-layer LSTM with $272, 102$ parameters on Sent140 following the architecture of Caldas et al. (2018). We use standard non-IID LEAF splits resulting in $59, 214$ train users and $39, 477$ test users and a local batch size of $32$. We train for $15$ epochs, and for $\varepsilon = 0$ achieve an average test accuracy of $64.8\%$. We train with a client learning rate of $\eta_C = 5.75$ and a server learning rate of $\eta_S = 0.244$.

- **Shakespeare** is a next character prediction task with $47$ classes (McMahan et al., 2017a). We train a similar LSTM model to Sent140, based on the architecture used by Caldas et al. (2018) with $819, 920$ parameters. We use standard non-IID LEAF splits with $1016$ train users and $113$ test users and a local batch size of $128$. We train our models for $15$ epochs resulting in an average final test accuracy of $44.4\%$ for $\varepsilon = 0$. We use a client learning rate of $\eta_C = 3$ and a server learning rate of $\eta_S = 0.524$.

# C    MEASURING EMPIRICAL PRIVACY

## C.1    PRIVACY MEASURES AND ACCOUNTING

Here we provide further details about privacy accounting and the different privacy measures that we analyse. In all experiments we use Rényi Differential Privacy (RDP) accounting with subsampling (Poisson sampling over the set of clients) to guarantee user-level DP. This is based on the DP-FEDAVG algorithm (McMahan et al., 2017b) with accounting implemented via the Opacus library (Yousefpour et al., 2021). More specifically, the accounting uses the RDP subsampling analysis derived from Mironov et al. (2019) and the RDP to $(\varepsilon, \delta)$-DP conversion from Balle et al. (2020).

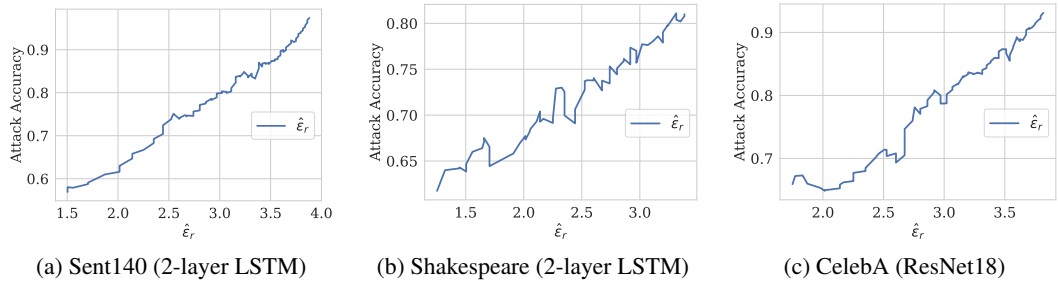

(a) Sent140 (2-layer LSTM)     (b) Shakespeare (2-layer LSTM)     (c) CelebA (ResNet18)

Figure 7: Per-round $\hat{\varepsilon}_r$ estimates against attack accuracy. Estimates are computed over 5 training runs with a final theoretical $\varepsilon = 50$.

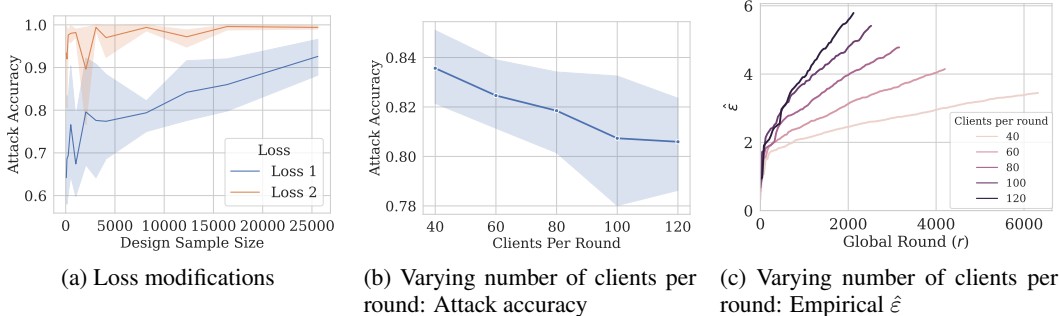

(a) Loss modifications     (b) Varying number of clients per round: Attack accuracy     (c) Varying number of clients per round: Empirical $\hat{\varepsilon}$

Figure 8: Further Ablations on CelebA; (a) CNN (b), (c); ResNet18

In this work, we have exactly four different privacy measures. For the theoretical quantities we have the per-round guarantee $\varepsilon_r$ and final privacy guarantee $\varepsilon$ which are computed as follows:

- $\varepsilon_r$ - This is the per-round theoretical epsilon that is derived from the RDP accountant when the subsampling rate $q$ is set to 1 and number of steps $R = 1$. This is a constant value (dependent on the noise multiplier $\sigma$ and $\delta$) and is the privacy guarantee of performing a single step of DP-FEDAVG.

- $\varepsilon$ - This is the theoretical $(\varepsilon, \delta)$-DP guarantee of the model trained under (user-level) DP. We calculate this via an RDP accountant with subsampling where the sampling rate $q$ is chosen to be the number of participating clients over the total number of clients in the population. Since we are using Gaussian noise this corresponds to the RDP analysis of the Subsampled Gaussian Mechanism (SGM), see (Mironov et al., 2019) for technical details.

Similarly, we have analogous empirical measures $\hat{\varepsilon}_r$ and $\hat{\varepsilon}$ which are computed as follows:

- $\hat{\epsilon}_r$ - Since the attack used by CANIFE infers membership of the canary update at a particular round, the privacy measure derived from a set of CANIFE attack scores is a per-round measure. This is computed via the formula derived in Kairouz et al. (2015) i.e.,

$$\hat{\varepsilon}_r = \max_{\gamma} \left( \log \frac{1 - \delta - \mathrm{FPR}_\gamma}{\mathrm{FNR}_\gamma}, \log \frac{1 - \delta - \mathrm{FNR}_\gamma}{\mathrm{FPR}_\gamma} \right),$$

where $\mathrm{FPR}, \mathrm{FNR}$ are computed from the attack scores at round $r$. In our experiments we maximise $\hat{\varepsilon}_r$ over the threshold $\gamma$ to provide a worst-case measure. The quantity $\hat{\varepsilon}_r$ is directly comparable to the theoretical $\varepsilon_r$. We also compute $95\%$ confidence intervals (CIs) for $\hat{\varepsilon}_r$ from the attack scores via the Clopper-Pearson method as in Nasr et al. (2021).

- $\hat{\varepsilon}$ - This is the empirical privacy measure of the model derived from CANIFE. One could apply basic composition to $\hat{\varepsilon}_r$ over $R$ rounds to obtain the empirical measure of $R\hat{\varepsilon}_r$ but this results in suboptimal composition. Instead we compute $\hat{\varepsilon}$ under the tighter composition

| Canary Sample after optimization |
|---|
| mRnt,,,,,,,,,,’d,,,,,R,,,,,A,,,,,,,,,,,,,A,,,,,,,,,,,,,,,,AA,,,,V2,E >H &3 4i |
| Yet I confess that often ere thisMday,x?h\nnPImh!v7DhNAd}I!H’;kXXI’PmP1Iert 6Fa |
| th, nothing bu; aZ empty box, sir, wQich in my lord’s beealf I cote to e? rtat 3 |

Table 1: Canary Samples on Shakespeare.

of RDP with amplification by subsampling. To do so, we convert each $\hat{\varepsilon}_r$ into an equivalent noise multiplier $\hat{\sigma}_r$ and compound the noise over a number of rounds with the accountant. The quantity $\hat{\varepsilon}$ is directly comparable to $\varepsilon$. See Section C.2 for more information.

## C.2 ALGORITHM FOR $\hat{\varepsilon}$

In Section 3.4, we explained how we obtain a per-round privacy measurement $\hat{\varepsilon}_r$ from CANIFE and compound this to form a global privacy estimate $\hat{\varepsilon}$ over a training run. We detail this method in Algorithm 2. In order to compound our per-round estimates $\hat{\varepsilon}_r$ from CANIFE, since we only attack the model every $s$ rounds we assume that the noise estimate $\hat{\sigma}_r$ remains constant between rounds $r$ and $r + s$ before we attack the model again and re-estimate the noise $\hat{\sigma}_r$.

## C.3 DETAILS FOR MONITORING PRIVACY

In Section 4.2, we present experiments using CANIFE to measure empirical privacy during the training run of federated models. We ran each training run five times and examples of these runs are displayed in Figure 4. Here we provide extra details of the training setup:

- **CelebA.** We train a ResNet18 model for 30 epochs and have 100 clients participate per training round. This results in 85 rounds per epoch. We freeze and attack the model every $s = 40$ rounds resulting in 64 empirical privacy estimates ($\hat{\varepsilon}_r$) across training.
- **Sent140.** We train for 15 epochs and have 100 clients participate per training round. This results in 593 rounds per epoch and 8895 rounds in total. We freeze and attack the model every $s = 100$ resulting in 90 empirical privacy measurements across training.
- **Shakespeare.** We train for 30 epochs and have 60 clients participate per training round. This results in 17 rounds per epoch and 510 training rounds in total. We freeze and attack the model every $s = 8$ rounds resulting in 64 empirical privacy estimates across training.

## C.4 RELATIONSHIP BETWEEN $\hat{\varepsilon}_r$ AND ATTACK ACCURACY

We display the relationship between the per-round measurement $\hat{\varepsilon}_r$ and the accuracy of the attack for models with a final privacy of $\varepsilon = 50$ in Figure 7. We also found a consistent relationship for $\varepsilon \in \{10, 30\}$.

## C.5 FURTHER EXPERIMENTS: SHAKESPEARE

In Figure 4, we displayed example per-round estimates $\hat{\varepsilon}_r$ across training runs for CelebA and Sent140. In Figure 6a we display a similar plot but for an example training run on Shakespeare. We also note that one can use CANIFE to measure $\hat{\varepsilon}_r$ for models that are trained without privacy. In Figure 6b, we show an example training run on Shakespeare without DP ($\varepsilon = \infty$). As mentioned in Section 3.4, the number of attack trials determines an upper bound on $\hat{\varepsilon}_r$. Here we can see that the empirical privacy measure is essentially constant throughout (non-private) training and consistently reaches these bounds. We note that $\hat{\varepsilon}_r = \infty$ only once (when the maximum attack accuracy reaches 100%).

## D FURTHER ABLATION STUDIES

**Loss modification.** As discussed in Appendix A, we can minimise the dot-product of the canary with the average model update ("Loss 1") or the covariance loss ("Loss 2") which we choose to

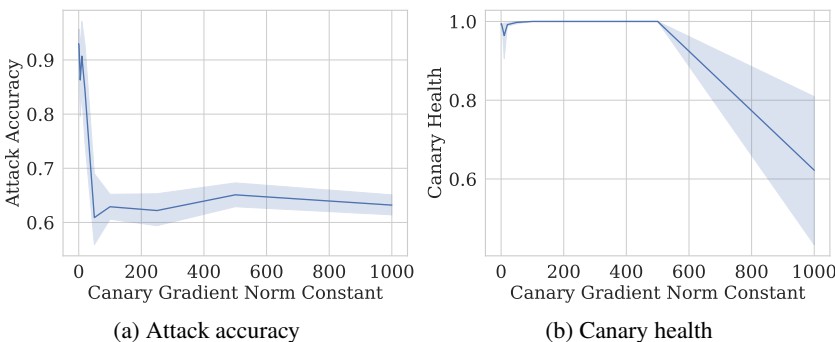

(a) Attack accuracy            (b) Canary health

Figure 9: Varying the canary gradient norm constant on CelebA (CNN)

use in all experiments. In Figure 8a, we vary the design sample size and carry out 5 attacks with a checkpointed CNN model on CelebA while varying these losses. We observe that there is a clear difference in accuracy between the two losses, with Loss 2 having consistently high attack accuracy which is almost constant as we increase the sample size. We note that Loss 1 seems more sensitive to the total number of design samples, with increasing attack accuracy as the design sample size increases.

**Number of clients per round.** In Figure 8b, we vary the number of clients per round and plot the average attack accuracy over 5 training runs. We note that the attack accuracy decreases slowly as the number of clients increases which is consistent with the observations made in Figure 3c. In Figure 8c, we plot the empirical estimate $\hat{\varepsilon}$ during training. We find that although the average accuracy decreases, this does not have a significant effect on the final estimate $\hat{\varepsilon}$ which decreases as the number of clients per round decreases.

**Canary gradient norm.** To conclude, we investigate the effect of the canary gradient norm constant in the second term of the loss $\mathcal{L}(z)$. We fix the privacy clipping constant to be $C = 1$ and vary the gradient norm constant on CelebA. We use a checkpointed CNN model trained without DP to $70\%$ test accuracy. We attack the model 10 times and average the results. In Figure 9a, we display the average accuracy of the attack and in Figure 9b the average canary health as we vary the gradient norm constant. We observe that choosing the constant too large can significantly decrease the efficacy of the attack and that choosing the constant around $C$ is enough to guarantee high accuracy and well-behaved optimization. We note that the canary health does not significantly decrease until the norm constant is chosen to be very large ($> 600$), yet the attack suffers a large drop in accuracy for constants $> 50$. This implies that as you increase the constant, it is first possible to design a canary that has large gradient norm but at the expense of orthogonality to the other model updates, and as the constant gets very large, it becomes too difficult to optimise either term of $\mathcal{L}(z)$.

## E  CANARY SAMPLES

We display in Table 1 some examples of designed canaries with CANIFE when initializing the crafting procedure with training sentences on Shakespeare.

## F  LIMITATIONS AND EXTENSIONS

In our experiments we have made assumptions for CANIFE that may be limiting in specific practical FL scenarios. In this section, we highlight such limitations with various extensions for CANIFE.

**Multiple Local Epochs.** Throughout Section 4, we have restricted clients to performing a single local epoch for simplicity. But in practice clients often perform multiple local epochs and this has shown to help improve the convergence of models trained in FL. CANIFE is not limited to methods

with a single local epoch and in practice to audit methods like DP-FEDAVG with multiple local epochs there are two main solutions:

1. **Use CANIFE as is:** There is nothing in the current formulation of CANIFE that does not apply to multiple local updates. Since in practice, the server will mock the adversary, they are free to choose how many local epochs the canary client performs. Thus, they can design a canary with a single local epoch to be orthogonal to clients who do multiple local epochs. We also believe this is most practical for an adversary since it is the easiest from the attackers viewpoint to design and optimize for.

2. **Modify CANIFE loss:** An alternative approach is to design a canary sample that has a model update (formed from multiple local epochs) that is orthogonal to all other model updates (which are also formed from multiple local epochs). To carry out such a design we can modify the CANIFE loss in equation 1 to include the canary model update $u_c$ instead of the canary gradient $\nabla_\theta \ell(z)$. In order to calculate the gradient of such a loss it requires backpropagating through the multiple local updates and essentially "unrolling" the local SGD steps. This may be computationally burdensome, and so while it is possible to design such a canary sample, it may not be practically viable for any adversary (and/or server) depending on the model size and number of local epochs. Thus in practice, it may be simpler to audit the model via (1).

**Multiple Privacy Measurements.** One current limitation of CANIFE is that it only produces a single privacy measurement ($\hat{\varepsilon}_r$) at a specific round. In practice, model auditors may want a more comprehensive empirical analysis of the model's privacy via multiple measurements, such as attacks that vary the threat model like that of Nasr et al. (2021). We believe CANIFE can be extended to support a "multiple measurement" approach with ease since there is some degree of freedom in the canary design. For example, by using different design pools, each one strengthening the adversary further (e.g. with more data and/or design pools that better approximate the federated distribution based on prior knowledge) and thus obtain a more holistic measure of the model's privacy. One can similarly vary the number of design epochs to simulate clients with limited computation. Designing multiple canaries per round under different constraints will generate a set of empirical epsilons that allow for more fine-grained statistics (for example, taking the maximum of the per-round empirical epsilons for a worst-case measure).

**Preventing Wasted FL Rounds.** Another limitation of CANIFE is that in our experimental setup we freeze the model for a set number of rounds to compute attack scores of a designed canary. In practice, the server and clients would be unwilling to waste federated rounds to compute CANIFE scores. We believe CANIFE can be extended to support a more practical attack without wasting training rounds as follows:

- **Multiple Canaries:** One alternative is that the server could design multiple canaries at a single round, and use these to obtain (multiple) attack scores (subject to compute limitations). This can help reduce the number of mock rounds being run.

- **Running measurements:** The previous approach may be computationally prohibitive depending on server resources and still requires frozen rounds. An alternative could be to maintain "running" attack measurements where we allow CANIFE to run alongside normal model training, letting the model change at each round. In this setup, the server can design a canary at each round, calculate attack scores and then proceed with updating the global model (without the canary inserted). This has minimal overhead to the server (who just needs to design the canary) and no additional overhead to clients (who just believe they are participating in a standard FL training round). The set of attack scores can be used to calculate empirical epsilons over various periods of training and one can change this period however they like. We emphasize that since our results (specifically Figure 4) show the per-round empirical measure is fairly stable after the first epoch or so of training, then the approach described here should still give stable results (even though the model would be changing at each round).

**Design Pool Assumptions.** The design pool is an important component of our attack, and in order to present a conservative (worst-case) privacy measure in this work, we assume that the adversary

has access to held-out data that approximates the federated training data well. To do this, we form the design pool from the test set of our datasets but in many practical scenarios this is not possible since the server may not know much about the (private) federated data. While this is a limitation of CANIFE, we believe that for many tasks it would be possible to form a design pool from public data. For example, if the adversary knows the exact task of the model (e.g. sentiment analysis) then the adversary can form a design pool from public datasets (e.g. Sent140) or even craft their own language data for canary design. We expect that this would be a reasonable proxy for the true federated dataset and would not significantly affect privacy measurement. In scenarios where the server has no prior knowledge, they could utilise private synthetic data generation methods or Federated Analytics (FA) to privately compute statistics about client data to guide the choice of design pool.

