# OpenReview forum: "CANIFE: Crafting Canaries for Empirical Privacy Measurement in Federated Learning"
_ICLR.cc/2023/Conference — ICLR 2023 poster_

### Official Review · Reviewer_K314 · 2022-10-24

**Confidence:** 4
**Correctness:** 4
**Technical Novelty And Significance:** 3
**Empirical Novelty And Significance:** 3
**Recommendation:** 8

**Clarity, Quality, Novelty And Reproducibility:**

The clairity of this paper is good in terms of explaining the threat model and the algorithms, and the experiments are well-executed. The proposed CANIFE is original. Though there is no link to the code, the implementation should be straightforward from the pseudo code and all necessary hyperparameters are provided and I believe the results should be reproducible.


**Strength And Weaknesses:**

Strength

1. The paper is well-motivated as currently training large models with DP has non-negligible impact on utility and it is less understood whether such strong guarantees are necessary under more realistic settings. This paper could be useful for developing more practical private learning algorithms.
2. The paper is well-written and easy to follow, and the proposed algorithms are straightforward and simple to implement.
The evaluation is done on multiple image and text datasets demonstrating the effectiveness of the proposed approach.

Weakness

1. Although the author mentioned a few existing empirical privacy loss estimation approaches, there is no empirical comparison between this work and prior works. For example, would any of the settings in Milad et al [1] be comparable with CANIFE, in terms of privacy loss estimation and attack efficiency?
2. The main goal of the paper is to monitor empirical privacy during FL training while one limitation is that one has to have a good Design Pool (i.e. held-out data in the experiments) to simulate the other clients for a more accurate measurement. If the design pool is very different from the real distribution, then the measurement could be overly optimistic. However, in many real FL applications, there is no held-out data available since these are also sensitive data on device, and having an accurate design pool might be hard to achieve. The paper would be stronger if the authors also provided methods to generate a design pool when no held-out data are available yet the privacy measurement is tight. Relatedly, the distribution assumption about the design pool is also not rigorously defined.

References:

[1] Milad Nasr, Shuang Songi, Abhradeep Thakurta, Nicolas Papemoti, and Nicholas Carlin. Adversary instantiation: Lower bounds for differentially private machine learning. In 2021 IEEE Symposium on Security and Privacy (SP), pp. 866–882. IEEE, 2021.


**Summary Of The Paper:**

This paper proposed CANIFE for measuring the empirical privacy loss under looser assumptions about the adversary’s capability compared to differential privacy (DP). CANIFE generates worst-case data samples in federated learning (FL) such that the gradients of such samples are orthogonal to most of the model updates from other participants, and from which the adversary can tell whether such outliers participated in training or not in each round. The authors evaluated the effectiveness of CANIFE on benchmark image and text datasets and showed that DP overestimated the privacy by 2 to 7 times.


**Summary Of The Review:**

Overall I think it is a good paper despite the weakness mentioned above.

---

> ### Author Response · Authors · 2022-11-17
> **Response to Reviewer K314**
>
> We would like to thank the reviewer for their positive comments and take the time to reply to some of their concerns.
>
> > **“(...) there is no link to the code (...)”**
>
> In order to aid the reproducibility of our work we have open-sourced our code anonymously with replication instructions at https://anonymous.4open.science/r/canife/ . We have added a corresponding footnote to our revised manuscript.
>
> > **“would any of the settings in Milad et al [1] be comparable with CANIFE, in terms of privacy loss estimation and attack efficiency?”**
>
> This is a great question. To be clear, the privacy loss estimation, (i.e., formula for empirical epsilon) is the same as that used by Milad et al and is the formula derived in [KOV15].
>
> Milad et al. do propose some attacks that can be applied to the Federated setting. For most attacks, they primarily assume that the adversary can directly modify their model updates to poison the model. In CANIFE, we assume that the adversary cannot directly tamper with model gradients or model updates (which can be enforced in practice with secure sandboxing) and instead must produce a real data example (the canary) that has the desired attack properties (good separation of two Gaussians).
>
> The closest attack to our setting is that of “Crafter 3” (or the “Adaptive Poison Input” attack) where they perform gradient ascent to construct a malicious sample (and do not modify gradients). But, as stated by the authors, this attack assumes “strong control by the adversary on the dataset” and its purpose is only for “establishing a lower bound” and is not a practical attack.  Instead, CANIFE considers a way to design such a malicious sample that doesn’t have such strong assumptions and is specifically designed for FL. This directly translates into a more realistic measure of the model’s privacy in practical FL scenarios under our realistic (yet conservative) threat model.
>
> On the other hand, while it is possible to use CANIFE in the central setting, as we show in Section 4.1 (Figure 3), our attack is essentially tight in an FL setting that corresponds to central training (each client holds a single example) and we don’t believe a comparison to central privacy measurement methods would be useful here since they also conclude similar things.
>
> *[KOV15] Kairouz, Peter, Sewoong Oh, and Pramod Viswanath. "The composition theorem for differential privacy." International conference on machine learning. PMLR, 2015.*
>
> > **“(...) the distribution assumption about the design pool is also not rigorously defined”**
>
> In all experiments we form the design pool from the test set of our datasets. As we utilise LEAF benchmarks, this test set is already split into non-IID clients which we take to be mock clients for use in the canary design. Thus, our experiments present the most conservative measure of privacy, i.e., the worst-case where an adversary has access to data that closely approximates the true training data. We have made this clearer in a modification to our manuscript.
>
> However, in general, we do not place formal assumptions on the design pool, and as mentioned in our response to Reviewer hrNj, the design pool may be useful to vary in practice to collect “multiple measurements”. For example, the server may vary the design pool to mimic multiple adversaries of different attack strengths to obtain multiple empirical measures and thus a more holistic view of the model’s privacy.
>
> > **“If the design pool is very different from the real distribution, then the measurement could be overly optimistic.”**
>
> > **“in many real FL applications, there is no held-out data available since these are also sensitive data on device, and having an accurate design pool might be hard to achieve.”**
>
> We agree that the design pool is an important component of our attack but do not believe it is such a significant assumption in practice. In many tasks it would be possible to form a design pool from public data, i.e, if the adversary knows the exact task of the model for example, sentiment analysis, then the adversary can form a design pool from public datasets (like Sent140) or even craft their own language data for canary design. We expect that this would be a reasonable proxy for the true federated dataset and would not significantly affect privacy measurement.
>
> > **“The paper would be stronger if the authors also provided methods to generate a design pool when no held-out data are available yet the privacy measurement is tight.”**
>
> We believe there are practical work-arounds to this such as private FL synthetic data generators or using Federated Analytics (FA) to privately compute distribution statistics to guide the choice of design pool. We do think this more general problem of generating a design pool in situations where information about the federated dataset is completely unknown is an interesting problem but is not the focus of our work.

---

### Official Review · Reviewer_hrNj · 2022-10-24

**Confidence:** 3
**Correctness:** 4
**Technical Novelty And Significance:** 4
**Empirical Novelty And Significance:** 3
**Recommendation:** 8

**Clarity, Quality, Novelty And Reproducibility:**

#Clarity and reproducibility

As mentioned under Strengths, the paper gives an excellent account of the method's motivation, implementation details and choices (paired with analytic experiments). Experiments are documented thoroughly. Standard software libraries and datasets are used to implement the experiments, which supports reproducibility.

#Novelty

The approach is novel in that it brings in the idea of the canary, known from different settings, to carry out a task in the DP-FedAvg setting, empirical privacy determination. This task is usually achieved quite differently.

#Quality

The work is of high quality. Many non-trivial technical steps had to be solved for the overall method to work. The experimental design is very complete and addresses all necessary aspects. The analysis is accurate, informed, state-of-the-art.

**Strength And Weaknesses:**

#Strengths

The privacy estimate provided by this method is practically useful as complement to a theoretical guarantee. The method is computationally accessible. The threat model is applicable in practice (though it looks to me as though client samples are wasted during training rounds ran for the purpose of the method).

The paper is remarkably well written, well structured, self-contained but with useful extra detail in the appendices. Experimental details, including other software libraries, datasets, other methods are detailed. Plots are clear, legible, to the point. Notation, nomenclature, external references are all unambiguous and helpful.

#Weaknesses

The proposed method seems to be limited intrinsically, probably like other methods with the same purpose: it requires a very specific protocol to be applied, and will return the specific privacy measurement corresponding to its threat model. However, complete practical privacy measurements presumably require multiple such methods, under varied threat models eg Nasr+ 2021, to form a complete picture as a counterpoint to the theoretical DP guarantee.

The threat model might be demanding, especially the availability of public data (design pool) and cooperative clients and servers.

Experiments are limited to classification tasks.

Looking hard for weaknesses, one could wish for the paper to be more overt about the method's shortcomings and limitations.

**Summary Of The Paper:**

Motivated by the insight that DP budgets are conservative (i.e. a given budget overestimates the privacy/membership inference information that can be extracted realistically), this work designs a canary-based method to produce a more realistic estimate of the privacy budget in FL settings. The threat model is
- safe but cooperative server
- one rogue client
- access to public model weights once trained
- access to a public dataset ("design pool")
- access to public noisy model updates

The proposed method injects (or not) a canary data point in a given training round, computes scores using public noisy model updates, then aggregates per-round privacy estimates to global privacy estimates. The canary is obtained by a computationally cheap local SGD relying on the derivative of the loss wrt the canary value, holding model parameters fixed.

**Summary Of The Review:**

The paper is very good and has very few flaws, if any. It addresses a problem of practical importance, comes up with a relatively simple, general solution which solves the problem in many cases. Therefore its impact can be expected to be strong.

---

> ### Author Response · Authors · 2022-11-17
> **Response to Reviewer hrNj (1/2)**
>
> We would like to thank the reviewer for their positive comments. We would like to clarify some of the points made.
>
> > **“(...) it looks to me as though client samples are wasted during training rounds ran for the purpose of the method”**
>
> This is a good point but one which we do not believe is too restrictive. For simplicity, in our experiments, we carry out the attack over multiple rounds where the model is frozen to obtain our privacy measurement, and as correctly pointed out this would waste training rounds which the server and clients would like to avoid in practice.
>
> We believe CANIFE can be extended to support a more practical attack without wasting training rounds. One alternative is that the server could design multiple canaries at a single round, and use these to obtain (multiple) attack scores which can help reduce the number of mock rounds being run.
>
> The above approach may be computationally prohibitive depending on server resources and still requires mock rounds. Another approach could be to maintain “running” attack measurements where we allow CANIFE to run alongside normal model training, letting the model change at each round. In this setup, the server can design a canary at each round and use it to calculate attack scores and proceed with updating the global model (without the canary inserted). This has minimal overhead to the server (who just needs to design the canary) and no additional overhead to clients (who just believe they are participating in a standard FL training round).
>
> The set of attack scores can be used to calculate empirical epsilons over various periods of training. For example, you could group attack scores every 50 rounds to produce empirical epsilons and one can change this period however they like to get more/less frequent measurements. We emphasize that since our results (specifically Figure 4) show the per-round empirical measure is fairly stable after the first epoch or so of training, then the approach described above should still give stable results (even though the model would be changing at each round).
>
> We have added this discussion to Appendix F in the newly revised manuscript.
>
> > **“The proposed method seems to be limited intrinsically, probably like other methods with the same purpose: it requires a very specific protocol to be applied, and will return the specific privacy measurement corresponding to its threat model. However, complete practical privacy measurements presumably require multiple such methods, under varied threat models eg Nasr+ 2021, to form a complete picture as a counterpoint to the theoretical DP guarantee.”**
>
> First, with regards to Nasr et al., our proposed CANIFE method addresses a specific gap in their proposed set of attacks in the sense that our method is focused on concrete FL use cases. Instead, the work of Nasr et al. tackles the academic question of whether DP is tight and in fact some of the attacks considered by them would be not practical to use for privacy auditing in FL.
> We believe CANIFE can be easily extended to support a “multiple measurement” approach since we have some degree of freedom in the canary design. For example, by using different design pools, each one strengthening the adversary further (e.g. with more data and/or design pools that better approximate the federated distribution based on prior knowledge) and thus obtain a more holistic measure of the model’s privacy. One can similarly vary the number of design epochs to simulate clients with limited computation. Designing multiple canaries per round under different constraints will generate a set of empirical epsilons that allow for more fine-grained statistics (for example, you can take the maximum of the per-round empirical epsilons for a worst-case measure).
>
> We have added this discussion to Appendix F in the newly revised manuscript.

---

> > ### Author Response · Authors · 2022-11-17
> > **Response to Reviewer hrNj (2/2)**
> >
> > *(continuing from previous comment)*
> >
> > > **“The threat model might be demanding, especially the availability of public data (design pool) and cooperative clients and servers.”**
> >
> > We have taken care to make sure the threat model is aligned with what can occur in practice and do not believe the assumptions made are that demanding. First, for the design pool, as long as the adversary knows the model’s task then it should be possible to construct a suitable design pool from a public dataset. For example, for most NLP problems it should be fairly simple to construct a design pool from public data or even to create your own e.g., for sentiment analysis one could just use the federated Sent140 dataset to form mock design clients as a reasonable proxy. In practical settings where the server emulates an adversary, it is likely the server has a priori knowledge about the data being collected e.g. access to statistics about the data distribution that can be acquired privately through Federated Analytics (FA) or even have access to some small subset of data that is fairly representative (e.g. collected from “opt-in” users).
> >
> > Secondly, we have chosen to focus on an honest-but-curious setting which is a now standard assumption in DP-FL and something that is widely assumed in practical deployments (e.g., [Ram+20] actually assumes a trusted server). We do not believe this is too stringent since this setting currently aligns the most to practical deployments but agree that empirical FL privacy measurement under more adversarial threat models would be interesting future work.  We also emphasize that CANIFE incurs no additional overhead to participating clients since the CANIFE procedure can be performed on the server as post-processing.
> >
> > *[Ram+20] Ramaswamy, Swaroop, et al. "Training production language models without memorizing user data." arXiv preprint arXiv:2009.10031 (2020)*
> >
> > > **“Looking hard for weaknesses, one could wish for the paper to be more overt about the method's shortcomings and limitations.”**
> >
> > Taking into account all the reviewers comments we have added a limitations section to our manuscript, explaining various limitations and possible extensions of CANIFE in practice. Due to space constraints we present this discussion in Appendix F.

---

### Official Review · Reviewer_FybW · 2022-10-25

**Confidence:** 3
**Correctness:** 3
**Technical Novelty And Significance:** 2
**Empirical Novelty And Significance:** 4
**Recommendation:** 5

**Clarity, Quality, Novelty And Reproducibility:**

As mentioned above, I have some questions about the quality of the algorithm proposed in the paper as it pertains to multi-client-epoch algorithms, as well as the experimental comparisons made.

I think the motivation of the paper is clearly explained and the algorithm is easy to understand, but as mentioned above, there are some seemingly implicit assumptions and parts of the experimental setup that I was unable to fully understand by reading the paper.

As mentioned above, the assumptions the authors place on the adversary in the auditing scheme, and in turn the auditing problem setting, are I believe novel.

**Strength And Weaknesses:**

The main strength of the paper is the approach of designing the CANIFE algorithm. I felt the authors did a good job motivating the assumptions. The assumptions on the adversary do lie in a nice space where they are slightly more powerful than what is possible in practice, but still far-removed from an information theoretic adversary, and I haven't seen such a set of assumptions before. The assumption that one can only generate model updates from data points within the data domain rather than arbitrary model updates is especially a natural one. Furthermore, the method for designing canary data points rather than canary model updates is intuitive and elegant. Overall the method is lightweight and I think would be easy to apply in practice on a variety of federated model training algorithms. In addition, pending the caveats in the subsequent bullet points, I think the experiment results convey an interesting message, that a slightly powerful adversary still exhibits a reasonably large separation between its empirical privacy guarantees and the theoretical ones.

I think a main weakness of the paper is that it seems to assume both in the algorithm's motivation, design, and in the experiments that the federated model update computed by a client with a single data point would be a rescaling of the gradient of the model sent to the client on that data point. However, this isn't necessarily true if e.g. clients do multiple epochs of training over their local data (and if the canary is acting honestly given the data, which seems to be an important assumption in the paper). The ability for clients to do multiple passes over their dataset is a key feature of federated learning (e.g. if one looks at McMahan et al. 2016, Figure 3, one sees that multiple passes actually can improve the training process if communication costs are the primary bottleneck). Since the loss CANIFE optimizes to choose its canary and the discussion in the paper (including e.g. the experiments) are based on this assumption, while CANIFE could still be applied to auditing central training methods or simple federated algorithms like FedSGD or FedAvg restricted to clients doing 1 epoch (for which this assumption does hold), it's unclear to me that CANIFE is an appropriate method for auditing even FedAvg in general. One thing that confuses this point for me is that in the experiments section, "epoch" seems to refer to a pass over the clients rather than a pass by the clients over their dataset. So it's unclear if clients are using one or multiple epochs in the experiment. One could perhaps show that the update generated by multiple client epochs has a large dot product with the initial gradient, in which case perhaps having the initial gradient be orthogonal is a good heuristic choice.

Also, it seems the RDP accountant was used for the theoretical epsilons, which as the authors mention is not tight if subsampling is used (which I believe is the case in the experiments), so the gap between the empirical and theoretical epsilons could be tightened by using a tighter accounting scheme to lower the theoretical epsilon (One might argue that clients in practice do not appear uniformly at random, but in that case one should prefer an algorithm like DP-FTRL anyway, I think). I think it is unlikely that e.g., entire gap would be closed, but I think part of the message of the paper (empirical epsilons are much smaller than theoretical epsilons, even for an adversary that is stronger than is practical) is weakened since this message is vacuously true if the theoretical epsilon is far from tight.

**Summary Of The Paper:**

The paper introduces CANIFE, a method for auditing the empirical privacy guarantees of federated learning algorithms. In short, CANIFE does the following: It considers a canary client with a single data point that contributes in k "frozen" rounds, and doesn't contribute in k other  "frozen" rounds (frozen means the global model update is computed but not applied). The data point is chosen to be as out-of-distribution as possible, using a small number of held out examples from the data distribution; in particular, the authors propose a loss function over the data domain whose optimizers should be data points whose gradient is orthogonal to the gradients of the held out examples, and as large in norm as possible. One can then compute an "attack score", equal to the dot product of the aggregated round update and the canary client's computed update. Comparing the attack scores for each set of k rounds, one can look at a hypothesis test for detecting the canary based on thresholding the attack score, and compute a per-round empirical (epsilon, delta)-DP for all delta using the false positive/negative rates of this test, which can be used to estimate the privacy guarantee of the whole algorithm. The authors argue this is an effective empirical auditing scheme since the adversary is much less power than an information theoretic one (e.g., no knowledge of the other data samples participating in a round, can only generate adversarial data points rather than adversarial model updates), but still has more power than an adversary in practice (e.g., has access to held out examples, gets to participate in multiple rounds).

The authors perform experiments auditing federated model training using CANIFE. The authors show that for a simple CIFAR10 training setup where every client has one example, the noise multiplier corresponding to the difference between the distributions of attack scores is a very good estimate of the actual noise multiplier used in training. For more realistic setups, the authors consider the CelebA, Sent140, and Shakespeare tasks and compute empirical epsilon values using CANIFE. The authors compute empirical epsilons varying from 4-7 for theoretical epsilons of 10, 30, 50.

**Summary Of The Review:**

Overall, I lean towards rejecting the paper due to the aforementioned concerns. I think the paper carries a very nice message, that by using constrained adversaries we can get more reasonable practical epsilon estimates, and the problem of optimizing over data points in a canary is a very interesting one. But the fact that the paper seems tailored to single-client-epoch algorithms and uses a suboptimal accountant for its theoretical bounds makes it hard to discern if CANIFE is broadly applicable to a variety of federated learning algorithms, and how much of the gap in empirical and theoretical epsilons is due to a suboptimal accountant. So while I really like the ideas in the paper, it's hard to be confident in their level of impact. Furthermore, I think since the paper is breaching into a somewhat new space (so e.g., it is unclear if the problem setup will still be interesting in a few years, it's unclear how the empirical results would compare to other approaches), it needs to be held to a somewhat higher standard.

---

> ### Author Response · Authors · 2022-11-17
> **Response to Reviewer FybW (1/2)**
>
> We thank the reviewer for their detailed and constructive review. We would like to respond to the concerns raised.
>
> > **“I think a main weakness of the paper is that it seems to assume both in the algorithm's motivation, design, and in the experiments that the federated model update computed by a client with a single data point would be a rescaling of the gradient of the model sent to the client on that data point. This isn't necessarily true if e.g. clients do multiple epochs of training over their local data (and if the canary is acting honestly given the data, which seems to be an important assumption in the paper).”**
>
> > **“(...) it's unclear to me that CANIFE is an appropriate method for auditing even FedAvg in general”**
>
> We don’t believe that CANIFE is limited to methods with a single local epoch and we don’t think this is a significant weakness of our proposed method. To clarify, it is correct that our experiments only use a single local epoch for simplicity. However, in practice to audit methods like FedAvg with multiple local epochs there are two main solutions:
>
>
> 1. **Use CANIFE as is:** There is nothing in the current formulation of CANIFE that does not apply to multiple local updates. Since in practice, the server will mock the adversary, they are free to choose how many local epochs the canary client performs. Thus, they can design a canary client who carries out a single local epoch with gradient orthogonal to clients who do multiple local epochs. We also believe this is most practical for an adversary since it is the easiest from the attackers viewpoint to design and optimize for.
>
>
> 2. **Modify CANIFE loss:** An alternative approach is to design a canary sample that has a model update (formed from multiple local epochs) that is orthogonal to all other model updates (who do multiple local epochs). To carry out such a design we can modify the loss function in CANIFE to include the model update instead of the canary gradient. The reason why we omit this formulation is that it may be computationally burdensome to perform such a backprop through multiple (local) model updates in order to compute the loss. So while it is possible to design a canary sample which produces an orthogonal model update, it may not be practically viable for any adversary (and/or server) depending on the model size and the number of local epochs. Thus it is simpler to audit the model via (1).
>
> > **“One could perhaps show that the update generated by multiple client epochs has a large dot product with the initial gradient, in which case perhaps having the initial gradient be orthogonal is a good heuristic choice.”**
>
> This is an interesting remark and something that is likely to be true, especially if clients do a large number of local epochs. This idea can be thought of as in-between options (1) and (2) discussed above and could be a more efficient way to implement (2) in practice.
>
> > **“One thing that confuses this point for me is that in the experiments section, "epoch" seems to refer to a pass over the clients rather than a pass by the clients over their dataset. So it's unclear if clients are using one or multiple epochs in the experiment. “**
>
> This is correct - in the paper we refer to “epoch” to mean a global epoch (i.e, a whole pass of the full federated dataset in expectation). We only consider a single local epoch in our experiments. We have modified our manuscript to make this clearer. We have also added discussion about the extension of our method to multiple local epochs in Appendix F.
>
> > **“it seems the RDP accountant was used for the theoretical epsilons, which as the authors mention is not tight if subsampling is used (which I believe is the case in the experiments), so the gap between the empirical and theoretical epsilons could be tightened by using a tighter accounting scheme to lower the theoretical epsilon”**
>
>
> > **“use of a suboptimal accountant for its theoretical bounds…”**
>
> It is correct that we are using the RDP accountant with subsampling (and we have made this clearer in a modification to the manuscript). It is true that the gap from our attack could be closed by new theoretical accountants, and while progress on this front is good we have chosen to focus on providing tighter empirical epsilons in the federated setting. Heuristically, with the implementation of recent accounting advances such as [GLW21] we have observed 4-6% improvements in the theoretical epsilon and so our gaps (which are a magnitude larger) will still remain. If the reviewer can think of a more appropriate accountant we are happy to provide results.
>
> *[GLW21] Gopi, Sivakanth, Yin Tat Lee, and Lukas Wutschitz. "Numerical composition of differential privacy." Advances in Neural Information Processing Systems 34 (2021): 11631-11642.*

---

> > ### Author Response · Authors · 2022-11-17
> > **Response to Reviewer FybW (2/2)**
> >
> > *(continuing previous comment)*
> >
> > > **“I think part of the message of the paper (empirical epsilons are much smaller than theoretical epsilons, even for an adversary that is stronger than is practical) is weakened since this message is vacuously true if the theoretical epsilon is far from tight.”**
> >
> > While this point is very valid, it is also applicable to the entire line of literature on empirical privacy measurement in the central setting, where depending on the threat model assumed, various gaps have been shown between the “theoretical” epsilon (i.e, whatever is calculated by the current SOTA accountant) and the empirical measurement derived from an attack.
> >
> > It perhaps helps to separate our contributions more concretely. We have demonstrated a significant gap between the current theoretical privacy accounting (subsampled Gaussian mechanism under RDP) and an empirical measure derived under CANIFE. This shows that in order to tighten the gap (theoretically) we have to make certain practical assumptions about the adversary that differ from what can be assumed in the central DP setting. While it is true that such a gap could shrink as theoretical accounting tightens, the improvement of the theoretical epsilon is likely to be in the order of percentage points whereas our gaps are an order of magnitude larger and thus believe this is a worthy contribution that will hold in the near future.
> >
> > > **“(....) it is unclear if the problem setup will still be interesting in a few years"**
> >
> > CANIFE is a general method that can be used to audit privacy in the FL setting. The main quantity that CANIFE measures is the “natural noise” created by the interference with other client’s model updates alongside the DP noise. This is not something that is trivial to model theoretically so we expect this work to stay relevant in the near future. Furthermore, even if tighter FL accounting methods become available, our attack can still be used to demonstrate whether significant gaps still exist. CANIFE also provides a simple and lightweight method that current deployments of FL can adopt to audit models. This contribution (i.e., a general method) is what should be considered for “future impact” and not so much the current 2-7x privacy gap.
> >
> > > **“...there are some seemingly implicit assumptions and parts of the experimental setup that I was unable to fully understand by reading the paper.”**
> >
> > We hope that our discussion here and our modifications to the manuscript have made the experimental setup clearer (e.g. use of subsampling, privacy accounting details, single local epoch, meaning of “epoch”).

---

### Official Review · Reviewer_rAJ2 · 2022-10-25

**Confidence:** 3
**Correctness:** 3
**Technical Novelty And Significance:** 2
**Empirical Novelty And Significance:** 2
**Recommendation:** 5

**Clarity, Quality, Novelty And Reproducibility:**

The approach is presented well.
It would help to show more details on how real/theoretical epsilon is computed. The authors mention that they compute RDP over rounds but it requires more details to show how shuffling and sampling is accounted for compared to FedAvg with DP work.
It would be also good to explain why previous works that estimate epsilon empirically cannot be used in Federated setting. And similarly can the canary approach suggested here be used for centralised setting?

**Strength And Weaknesses:**

Strengths:
- it seems to be one of the first works to look into empirical epsilon estimate in FL
- the method is intuitive and based on examples that would cause worst case clipping

Weaknesses:
- DP guarantees are not explained well (i.e., is epsilon per user or overall epsilon, how doe shuffling come into play)
- it is not clear how the work is different from that of empirical estimation of epsilon in central setting (i.e., previous work in this space)

**Summary Of The Paper:**

The paper proposes an empirical evaluation of differential privacy parameter epsilon in federated learning setting. It does so by inserting canaries and seeing if it can determine the presence of these canaries in the model after each round update. It then uses this to estimate per round epsilon.

**Summary Of The Review:**

The paper considers an important problem but it is unclear how it is different from previous work on estimating epsilon in centralised setting and where those methods would fail.

---

> ### Author Response · Authors · 2022-11-17
> **Response to Reviewer rAj2 (1/2)**
>
> We thank the reviewer for their detailed and constructive comments and would like to address the concerns raised.
>
> > **“It would help to show more details on how real/theoretical epsilon is computed.”**
>
> When training models with DP, we use Rényi Differential Privacy (RDP) [Mir17] with subsampling for a fixed privacy budget. We utilise the RDP accounting implemented via the Opacus library [You+21] which uses the subsampling analysis derived from [MTZ19] and the RDP to $(\varepsilon,\delta)$-DP conversion from [Bal+20]. We have modified our manuscript to make this clearer.
>
> Note that $\varepsilon$ is only an upper bound of the “true” epsilon of the model and is currently the tightest formal measurement of a model’s privacy guarantee available in the literature. Our attack provides such a lower bound on the “true” privacy budget and illustrates the gap between theory and practice.
>
> > **“DP guarantees are not explained well (i.e., is epsilon per user or overall epsilon, how doe shuffling come into play)”**
>
> > **“The authors mention that they compute RDP over rounds but it requires more details to show how shuffling and sampling is accounted for compared to FedAvg with DP work.”**
>
> We use standard RDP accounting with amplification by subsampling (as in [MTZ19]) over the population of clients to guarantee user-level differential privacy. Clients are chosen uniformly at random (Poisson sampling) to participate in an FL round. We have made changes in the manuscript to make this clearer in our experimental setup (start of section 4).
>
> The epsilons $\varepsilon, \hat\varepsilon$ are an overall user-level DP guarantee (i.e, the cumulative privacy budget across multiple rounds of training). The per-round measurements ($\varepsilon_r, \hat\varepsilon_r$) are also user-level DP guarantees but at a specific round of training.
>
> Based on this discussion we have added Appendix C.1 to help clarify to readers the different measures of privacy and the specifics of the accounting with references to key literature on user-level DP with subsampling in FL. We hope that this addresses the main confusion, but stress that our accounting is similar to the FedAvg with DP work and is based on standard RDP accounting in the literature. We are happy to clarify any further details.
>
> *[Mir17] Mironov, Ilya. "Rényi differential privacy." 2017 IEEE 30th computer security foundations symposium (CSF). IEEE, 2017.*
>
> *[You+21] Yousefpour, Ashkan, et al. "Opacus: User-friendly differential privacy library in PyTorch." arXiv preprint arXiv:2109.12298(2021).*
>
> *[MTZ19] Mironov, Ilya, Kunal Talwar, and Li Zhang. "Rényi differential privacy of the sampled gaussian mechanism." arXiv preprint arXiv:1908.10530 (2019).*
>
> *[Bal+20] Balle, Borja, et al. "Hypothesis testing interpretations and renyi differential privacy." International Conference on Artificial Intelligence and Statistics. PMLR, 2020.*
>
> *[McM+17] McMahan, H. Brendan, et al. "Learning differentially private recurrent language models." arXiv preprint arXiv:1710.06963(2017).*
>
> *[KOV15] Kairouz, Peter, Sewoong Oh, and Pramod Viswanath. "The composition theorem for differential privacy." International conference on machine learning. PMLR, 2015.*
>
>
> > **“can the canary approach suggested here be used for centralised setting?”**
>
> While CANIFE can be used in the central setting (as is), the threat model we propose is more realistic in the FL setup than in the central setup, where the adversary would typically only access the final model (model release) instead of all intermediate (batched) gradients. Additionally, CANIFE has been explicitly designed for a practical FL setting where the attack infers the presence of a user at a specific FL round and there is no direct equivalence in the central setting. CANIFE also puts further restrictions on the adversary (e.g. must produce a valid sample, must craft the canary on a design pool) which differs heavily from the assumed threat model in central setting attacks.
>
> Note that the experiments in Figure 3 show CANIFE in a setting that is closest to the central setting, where we consider a FL setup with each participant holding a single sample (and thus the number of clients corresponds to the notion of a batch size in the central setting). We conclude that the attack is tight in this setting and so direct comparison to central methods would not be that insightful.

---

> > ### Author Response · Authors · 2022-11-17
> > **Response to Reviewer rAj2 (2/2)**
> >
> > *(continuing from the previous comment)*
> >
> > > **“it is not clear how the work is different from that of empirical estimation of epsilon in central setting (i.e., previous work in this space)”**
> >
> > > **“It would be also good to explain why previous works that estimate epsilon empirically cannot be used in Federated setting.”**
> >
> > These are great points - the main difference between our attack in the FL setting and that of existing central DP attacks (apart from difference in training) is the assumed threat model. In the central DP setting, attacks typically assume the adversary knows the existence of other training points and/or allows for poisoning the model by modifying gradients. For example, the work of Nasr et al. primarily assumes the adversary can directly modify model updates to poison the model. In CANIFE, we assume the adversary cannot directly tamper with model gradients and must produce a real data example (the canary). We also assume the adversary has no knowledge of other data points in the training set and must design an attack based on held-out/public data. These assumptions are what separates us from previous (central DP) work as our threat model does not allow for direct modification to gradients/model updates and so most attacks are not directly comparable.
> >
> > *[Nasr+21] Milad Nasr, Shuang Songi, Abhradeep Thakurta, Nicolas Papemoti, and Nicholas Carlin. Adversary instantiation: Lower bounds for differentially private machine learning. In 2021 IEEE Symposium on Security and Privacy (SP), pp. 866–882. IEEE, 2021.*

---

### Author Response · Authors · 2022-11-17
**Response to all Reviewers**


We would like to thank all reviewers for their detailed and constructive reviews. We have responded to each reviewer individually to address any comments/concerns.

We have also updated the manuscript with the following changes (new changes are highlighted in blue):

* We have included a link to our (anonymous) open-source code with replication instructions (see https://anonymous.4open.science/r/canife/).
* Based on comments by all reviewers, we have added Appendix F which includes limitations and extensions of CANIFE that encompass concerns from reviewers about how to use CANIFE in practice (design pool assumptions, how to provide multiple privacy measurements, how to prevent wasted FL rounds etc.)
* We have added Appendix C.1 to provide more details about the privacy accounting and re-iterate/clarify any new terminology introduced in our paper (e.g. per-round epsilon vs global model epsilon)
* We have cleaned up various technical terminology and missing details throughout the paper (FL epoch vs local epochs, our design pool formulation, accounting method used etc.)
* We have fixed a few typos, most notably cleaning up some notation related to the exposition of the attack (in terms of separating Gaussians) in Section 3.2.

---

### Decision · Program_Chairs · 2023-01-20

**Decision:**

Accept: poster

**Justification For Why Not Higher Score:**

Well executed but not particularly novel ideas

**Justification For Why Not Lower Score:**

Strong support from two reviewers one of whom works in this area.

**Metareview: Summary, Strengths And Weaknesses:**

This paper designs a relatively strong way to audit the privacy property of training a model via SGD in FL. It shows how an adversary can design an example that will lead to a model update with certain properties aimed at maximizing the probability of detecting the update in the final model. They demonstrate that this auditing leads to privacy levels several times stronger than those implied by theoretical DP guarantees.  One of the new and useful aspects of this work is that it creates actual inputs to the model and not just the gradient.
This is useful progress in a line of work on auditing which aim to establish "empirical epsilon" (arguably, a rather misleading name). It's main weakness is that it is naturally geared toward a rather specific training scheme and specific distinguishing test.

**Note From Pc:**

if the above contains the word "oral" or "spotlight" please see: "oral" presentation means -> notable-top-5% and "spotlight" means -> notable-top-25%. As stated in our emails, we are disassociating presentation type from AC recommendations

**Summary Of Ac-Reviewer Meeting:**

It turned out to be infeasible to find a suitable time due to different time zones and busy schedules.